# Landscape of FLT3 Variations Associated with Structural and Functional Impact on Acute Myeloid Leukemia: A Computational Study

**DOI:** 10.3390/ijms25063419

**Published:** 2024-03-18

**Authors:** Zeenat Mirza, Dalal A. Al-Saedi, Nofe Alganmi, Sajjad Karim

**Affiliations:** 1King Fahd Medical Research Center, King Abdulaziz University, Jeddah 21589, Saudi Arabia; 2Department of Medical Laboratory Sciences, Faculty of Applied Medical Sciences, King Abdulaziz University, Jeddah 21589, Saudi Arabia; skarim1@kau.edu.sa; 3Department of Biochemistry, Faculty of Sciences, University of Tabuk, Tabuk 48322, Saudi Arabia; dalsaedi@ut.edu.sa; 4Center of Excellence in Genomic Medicine Research, King Abdulaziz University, Jeddah 21589, Saudi Arabia; nalghanimi@kau.edu.sa; 5Computer Science Department, Faculty of Computing and Information Technology, King Abdulaziz University, Jeddah 21589, Saudi Arabia

**Keywords:** FLT3, variant, acute myeloid leukemia, annotation, prediction, I836, sorafenib

## Abstract

Acute myeloid leukemia (AML) is hallmarked by the clonal proliferation of myeloid blasts. Mutations that result in the constitutive activation of the fms-like tyrosine kinase 3 (*FLT3*) gene, coding for a class III receptor tyrosine kinase, are significantly associated with this heterogeneous hematologic malignancy. The fms-related tyrosine kinase 3 ligand binds to the extracellular domain of the FLT3 receptor, inducing homodimer formation in the plasma membrane, leading to autophosphorylation and activation of apoptosis, proliferation, and differentiation of hematopoietic cells in bone marrow. In the present study, we evaluated the association of *FLT3* as a significant biomarker for AML and tried to comprehend the effects of specific variations on the FLT3 protein’s structure and function. We also examined the effects of I836 variants on binding affinity to sorafenib using molecular docking. We integrated multiple bioinformatics tools, databases, and resources such as OncoDB, UniProt, COSMIC, UALCAN, PyMOL, ProSA, Missense3D, InterProScan, SIFT, PolyPhen, and PredictSNP to annotate the structural, functional, and phenotypic impact of the known variations associated with *FLT3*. Twenty-nine *FLT3* variants were analyzed using in silico approaches such as DynaMut, CUPSAT, AutoDock, and Discovery Studio for their impact on protein stability, flexibility, function, and binding affinity. The OncoDB and UALCAN portals confirmed the association of *FLT3* gene expression and its mutational status with AML. A computational structural analysis of the deleterious variants of FLT3 revealed I863F mutants as destabilizers of the protein structure, possibly leading to functional changes. Many single-nucleotide variations in *FLT3* have an impact on its structure and function. Thus, the annotation of *FLT3* SNVs and the prediction of their deleterious pathogenic impact will facilitate an insight into the tumorigenesis process and guide experimental studies and clinical implications.

## 1. Introduction

Acute myeloid leukemia (AML) is a rare but aggressive and fatal type of cancer. The five-year survival rate for AML is 29.5% [1]. For people younger than 20, the five-year relative survival rate is ~70% [2,3]. Extended exposure to environmental carcinogens, existing co-morbidities, reduced tolerance to intensive therapies, and the accumulation of specific genetic mutations during cell division correlate with a higher incidence of AML and lower survival rates in the elderly population [4]. Moreover, relapsed or refractory patients who endured hematopoietic stem-cell transplantation have an increased 5-year probability of overall survival [5].

AML was earlier classified into subtypes using FAB nomenclature based mainly on how the leukemia cells looked under a microscope (morphological and cytochemical criteria), but presently, the AML classification system developed by the World Health Organization (WHO) mostly follows cell characteristics and genetic abnormalities and catalogs AML into several types [6]. In approximately two-thirds of AML cases, signaling and kinase pathway gene mutations (e.g., FLT3, KRAS, NRAS, PTPN11, NF1, and KIT) are the most common mutational subset in AML, causing atypical activation of cellular signaling pathways [7]. Other primary mutations having clinical implications involve genes *TP53*, *GATA2*, *NPM1*, *CEBPα*, *WT1*, *BAALC*, *ERG*, *MN1*, *DNMT*, *TET2*, *IDH1/2*, *ASXL1*, *RUNX1*, and *CBL* [7,8].

Pathogenic mutations in the fms-like tyrosine kinase 3 (*FLT3*) gene are linked with a poor prognosis and overexpressed on most AML blasts. Pathogenic mutations, specifically internal tandem duplication in the juxta-membrane region (exon 14, 15) and point mutations in TKD (exon 20, mainly D835: D835Y, D835V, D835H, D835E, D835N, followed by I836H, I836M, V592A, Y842C, etc.), in the FLT3 gene are linked with a poor prognosis and overexpressed on most AML blasts [9,10]. Both FLT3-ITD and FLT3-TKD mutations have shown prognostic significance in AML patients [11,12,13,14]. FLT3, a class III receptor tyrosine kinase family member expressed by immature hematopoietic cells, is necessary for the proper development of stem cells [15]. Its aliases are CD135, FLK2 (fetal liver kinase-2), and STK1 (human stem cell kinase-11). The structural elements of FLT3 include an immunoglobulin-like extracellular ligand-binding domain (253–343), a helical transmembrane domain (544–563), the intracellular module comprising a JM autoregulatory dimerization domain (591–597), a highly conserved intracellular kinase domain interrupted by a kinase insert (610–943), and a C-terminal tail [16,17]. Mutations resulting in the constitutive activation of this receptor result in AML and ALL. Leukemias are exemplified by the presence of an activating mutation of the FLT3 transmembrane tyrosine kinase, which is either an internal tandem duplication (ITD) in the JM region, seen in around 20% of AML cases and more frequently within younger adult patients, or the most frequent D835Y point mutation in the activation loop [18]. FLT3-ITD mutations are clearly linked with proliferative AML (e.g., higher WBC, leukocytosis, and blast %) and also a higher risk of relapse, implicating declined overall survival and a poor prognosis [19]. Other common D835 substitution mutations include D835V, D835H, D835E, and D835N, which occur independently of FLT3/ITD [14]. Moreover, a plethora of other SNVs have been reported to have a pathological impact on AML.

Over 30% of AML patients possess activating *FLT3* mutations, hence making it an attractive prognostic and therapeutic target [18]. Numerous reported variations and mutations in *FLT3* could potentially cause resistance against the developed FLT3 inhibitors [20]. Activating length mutations in the juxta-membrane (JM) domain of the *FLT3* gene and mutations in its catalytic domain (D835/836) correspond to the most frequent genetic variations in AML. A 6 bp insertion in exon 20 of the *FLT3* gene (p.Ser840_Asn841insGlySer) in AML has also been previously reported [21]. The major types of FLT3 resistance involve the ‘gatekeeper’ mutation F691L, which exhibits universal resistance against all the currently used inhibitors [22], and the activating KD mutations D835V/Y/F and Y842C/H, which induce resistance against type 2 inhibitors. Mutations N676K and F691L lead to resistance against midostaurin; F691L causes resistance to gilteritinib; F691L, D835F/V/Y, and Y842C/H cause resistance to quizartinib; F691L, D835F/V/Y, and Y842C/H lead to resistance against sorafenib; and F691L induces resistance to crenolanib [23]. The KD mutations Y842H and D835F kinetically stabilize the active conformation, and the M664I mutation reportedly increases the catalytic efficiency of the enzyme and leads to resistance against the inhibitor pexidartinib [24]. Hence, in AML patients resistant to specific inhibitors, a combination of multiple drugs is administered, and ongoing clinical trials suggest combining present therapies with hypomethylating agents like azacitidine or decitabine too [25].

Based on their potency and specificity in inhibiting FLT3, FLT3 inhibitors are categorized as either first-generation nonselective multi-kinase inhibitors (for example, midostaurin, sunitinib, lestaurtinib, sorafenib, and tandutinib) or second-generation selective inhibitors (for example, gilteritinib, crenolanib, and quizartinib) [26]. Additionally, based on the bound conformation of the tyrosine kinase domain (TKD), FLT3 inhibitors are categorized into type I (midostaurin/PKC412, lestaurtinib, gilteritinib, and crenolanib, which bind the active DFG-in conformation) and type II (sorafenib, quizartinib, and tandutinib, which bind the inactive DFG-out conformation) [27]. Second-type inhibitors are designed to be more potent and selective for FLT3. Treatment with sorafenib in combination with intensive chemotherapy may offer some potential benefits in terms of relapse-free and overall survival for previously untreated AML patients [28,29]. Several other novel kinase inhibitors have been developed for FLT3 inhibition, like G-749 [18,30]. Next-generation sequencing of cancer cells has facilitated the identification of millions of mutations and variations, yet not all the identified mutations in cancer genomes contribute to the initiation or progression of malignancy; the rest might alter cellular processes beyond oncogenesis. Mutations that yield a selective growth advantage, thereby promoting cancer growth, are named driver mutations, and those that do not are called passenger mutations. Identifying mutations that contribute to cancer inception is a fundamental step towards comprehending cancer biology for targeted oncotherapies [31]. Variations in the DNA sequences determine the disease susceptibility, development, advancement, and response to pathogens, allergens, chemicals, drugs, and vaccines. Single-nucleotide variation (SNV) annotation is imperative to predict the functional effect of each variation and is crucial for personalized therapies. The real reflection of any disease phenotype is seen at the protein structural level. The foremost cause of genetic diseases, including cancer, is a single-base DNA variant resulting in an amino acid substitution that can impact protein function by altering 3D shape and electrostatics, thereby either hampering the folding and stability of the polypeptide chain; affecting posttranslational modification sites, ligand binding, and catalytic activity; or interacting with binding partners [32].

FLT3 is a dominant oncogene, and any genetic alterations may be associated with structural differences and functional dynamics, rendering tumors sensitive or resistant to specific inhibitors. These point mutations might impose a subtle yet profound impact on conformation locally at the site but also on cooperative interactions and allosteric regulation in signaling pathways [33]. Since FLT3’s variation I836 is comparatively less studied than D835, we emphasized these variations and their impact on the interaction with sorafenib.

This study was designed to predict and understand how certain variants affect FLT3 protein function and determine whether any of these known variants may be correlated with either an increased or decreased cancer risk. Even though FLT3 kinase activity can be increased by the TKD mutation alone, it is still unclear how these factors relate to the pathophysiology and prognosis of AML. Hence, further investigation is needed. A systematic prioritization of disease-causing SNVs identified in *FLT3*, a key cancer biomarker gene, and the prediction of the functional effects of missense variants using knowledge-based learning methods were attempted. Numerous SNVs have been reported in the *FLT3* gene, but their thorough structural and functional annotation has not been reported. Hence, a comprehensive evaluation of FLT3 variations and their significant implications in the context of AML was attempted. This computationally intensive analysis offers a deeper holistic understanding and might open new avenues for targeted therapeutic strategies. Moreover, the methodology adopted in this study sets a new benchmark for variant annotation for innumerable disease biomarker genes, highlighting the importance of computational tools in the exploration of genetic variations and their clinical implications.

## 2. Results

### 2.1. Association of FLT3 with AML

Mutations resulting in the constitutive activation of this receptor are reportedly synonymous with AML, as confirmed via the UALCAN database (Figure 1A). The upregulation of *FLT3* positively correlates with AML (Figure 1B). FLT3 mutations in AML are associated with high FLT3 expression. An expression profile is an independent measurement of mRNA in case or control samples, while a differential expression requires expression values from both case and control samples to obtain a log2 fold change for comparative value. The RNA expression levels of FLT3 in normal vs AML cases using the OncoDB web portal confirm the association of the *FLT3* gene with AML. The OncoDB database shows expression values for AML only. However, FLT3’s transcript per million in AML (TPM = 342) is quite high compared to other cancers (TPM ranging 5–125). In general, the expression levels of genes can be derived from TPM values (no expression, TPM < 0.5; low expression, TPM = 0.5–11; medium expression, TPM = 11–1000; and high expression, TPM > 10,000). The average TPM for AML was 342, which indicates medium expression.

### 2.2. Retrieval of SNVs and Deleterious Variants

All the missense variants reported in dbSNP and ClinVar were listed. The majority of the 29 FLT3 variations listed in Table 1 were classified as substitution-missense; according to SIFT’s categorization, 27 of these variants had deleterious effects, while F594L, D651G, and I687F were anticipated to be tolerant. SIFT’s pathogenic predictions were confirmed by PolyPhen2 and PredictSNP. Except for V579A, V592A, I836L, I836V, and D839G, which were predicted to be neutral with a confident score of less than 85%, all the effects of AA substitutions projected by SIFT were consistent with the results from other tools. Amongst all the missense SNVs listed in Appendix A, nine are pathogenic (I836M/L and D835E/A/V/Y/N/H) and clinically significant for AML. A total of 26 are categorized as likely pathogenic, 10 as likely benign, and 7 as benign, while 21 have uncertain significance, and the rest are not provided. Variations at 835 and 836 lie in the catalytic domain of FLT3. The total number of distinct variants of FLT3 based on MOKCa was 297. Only selected FLT3 variations, along with their predicted effects and frequencies, are shown in Figure 2A,B.

### 2.3. Deleterious Variants and Their Effect on FLT3 Function

The potential effects of variants on FLT3 function are summarized in Table 2. A score of more than 0.75 for MutPred2 was associated with enabling disease, as indicated by SNPs&GO; however, Y591C, which had an impact score of 0.69, affected the inference of molecular pathways and was identified as pathogenic with high confidence. Variant I836S had a significant effect on FLT3 function, as projected by its MutPred2 score (0.89), and its pathogenicity, confirmed by SNPs&GO results, showed that it causes disease with high confidence (6 degrees). It was shown that most variations reported for D835 were associated with loss of relative solvent accessibility and loss of allosteric site at R834, and, thus, were projected as deleterious. Contrariwise, it was predicted that I836 variants gain relative solvent accessibility and gain the allosteric site at R834. As a result, their effects were classified as disease for I836F and I836S and neutral for I836L, I836M, and I836V.

### 2.4. Deleterious Variants and Their Effects on FLT3 Structure

The overall FLT3 structure illustrating the catalytic kinase domain is depicted in Figure 3. The selected mutational hotspots of FLT3 are shown in Figure 4 to exhibit their structural localization. The impact of the variants on FLT3’s structure and stability is illustrated in Table 3. In an analysis of Missense3D, most variants would not display any structural damage except for Y572C, V579A, F594L, G619C, K663Q, N676K, and I836F, which were shown to cause damage. I836F was significant, having computed damage in the cavity positioned in the DFG region (Figure 3 and Figure 4). Y591C, D835F, and D835V were projected to remain unaltered structures, and the stability of variations G619C, D835F, D835V, and I836F was found to be destabilizing. Although MutPred2 computed the effect of I836S on mechanism-changed stability, it did not predict the effect on stability.

### 2.5. Molecular Docking and Interaction Analysis

Wild-type FLT3 interacted with sorafenib with a binding affinity of −8.41 kcal/mol (Figure 5A), and the 3D structures of variants I836F, I836M, and I836V had identical types of interactions with sorafenib with binding affinities of −8.23, −7.79, and −8.48 kcal/mol, respectively (Figure 5B). It was noticed that these variants were stabilized by four H-bonds with Ser547, Gln577, Glu661, and Cys694 residues (Figure 6), and the remaining FLT3 structures only formed two hydrogen bonds. On the other hand, lower binding energy was formed when sorafenib interacted with I836L and I836S (−6.64 and −6.37 kcal/mol). Complex I836L had an SASA value of 20,090.4, which was higher than the value of the wild type (20,003.9), whereas I836V was characterized by a lower SASA value for both the protein and the complex.

## 3. Discussion

AML, being a heterogeneous, complex hematological malignancy, exhibits a broad mutational landscape [7,8]. Intriguingly, mutations within the *FLT3* gene are seen in approx. one-third of AML patients. *FLT3* is widely expressed in hematopoietic progenitor cells and is overexpressed in the majority (up to 50%) of AML blasts [9,10,34,35,36,37,38]. The *FLT3* gene translates into a class III RTK that controls hematopoiesis following its activation by binding the fms-related tyrosine kinase 3 ligand to the extracellular domain, inducing homodimer formation in the plasma membrane and causing autophosphorylation of the receptor. The activated RTK then phosphorylates and constitutively triggers numerous cytoplasmic effector molecules in several downstream signaling pathways, mainly in apoptosis, proliferation, and differentiation of hematopoietic cells in bone marrow, particularly the phosphatidylinositol 3 kinase PI3K/AKT prosurvival pathway and the RAS/RAF/MEK/ERK cascade [7,39]. Dysregulated glycogen synthase kinase-3 (GSK-3) is also implicated in the biology of AML [40], and GSK3 inhibitors typically decrease the viability of cells harboring FLT3-ITD mutations. Activating mutations of FLT3 constitutively activate β-catenin by inhibition of GSK-3β in a PI3 kinase pathway-dependent manner; in other words, the Wnt/β-catenin pathway controls the sensitivity of the mutant FLT3 receptor kinase inhibitors via a GSK-3β-dependent way. Hence, the potencies of the inhibitors of FLT3 kinase activity could be modulated by the activity of the Wnt/β-catenin pathway in cells having FLT3-ITD mutations. FLT3-ITDs signal through GSK-3β to activate β-catenin, perhaps directly contributing to the leukemic phenotype [41,42].

Upon activation, the FLT3 protein undergoes a conformational change involving the flipping of three conserved residues, Asp-Phe-Gly (DFG), in the activation loop; hence, there exist two conformational forms of FLT3: active (DFG-in) and inactive (DFG-out). Interestingly, type I inhibitors are active in cells with either FLT3-ITD or FLT3 KD point mutations, whereas type II inhibitors are active in cells having FLT3-ITD but not FLT3 KD point mutations [43]. All FLT3 inhibitors interact with the ATP-binding region of the intracellular TKD and competitively inhibit ATP binding, thereby prohibiting receptor autophosphorylation and activation of downstream cascades. On the other hand, type I inhibitors bind to the ATP-binding site when the receptor is active, while type II inhibitors interact with a hydrophobic region immediately adjacent to the ATP-binding site that is only accessible when the receptor is in its inactive form, thereby preventing receptor activation [43,44]. A molecular dynamics simulation study for variant drug responses due to the *FLT3* G697R mutation on PKC412 (type I) and sorafenib (type II) was conducted by Lee et al., wherein they showed that PKC412, being larger, uses its indolocarbazole lactam rings to occupy the adenine pocket, compared to the slender sorafenib, which stretches into both the adenine and back pockets [45].

To better understand FLT3 signaling and develop effective therapeutic strategies for FLT3-driven cancers, research on DFG-out is crucial. Hence, the DFG-out structure 1RJB was strategically used as a model to investigate FLT3 variations and inhibitor interactions. Based on the annotated variants of the FLT3 protein, we found that D835 and I836 were more deleterious, having the potential to influence molecular pathways associated with important roles for this receptor, including altered transmembrane protein and changed stability, all of which showed a high degree of confidence in their capacity to disrupt kinase signaling and induce pathogenesis. Conserved aspartate in the activation loop is an oncogenic hotspot found not only in the case of FLT3-D835 but also similarly in other kinases, for instance, MET-D1228, KIT-D816, and PDGFRa-D842 [33]. A systematic evaluation of FLT3 mutations and their effects on AML was undertaken using computational approaches. FLT3-D835 is a structurally conserved position and is also a known driver mutation site. The D835 missense substitutions D835A/E/F/H/N/V/Y are commonly occurring activating mutations predicted to have deleterious and pathogenic effects (Table 1 and Table 2), but they are predicted by Missense3D to have no structural damage (Table 3). D835V causes structural change, as indicated by the unwinding of the critical 3_10_ helix, which enhances local protein mobility and destabilizes the autoinhibited kinase conformation [33].

Molecular docking helps in gaining vital insights into how genetic variants impact inhibitors’ binding affinity and effectiveness by assessing the differential interaction between drugs and wild-type and mutant protein structures. The development of more potent therapies for genetic diseases and customized medicine strategies depends on this knowledge. We tried to shed light on the structural underpinnings of sorafenib’s binding to FLT3 and evaluate the potential effects of certain variants on the binding affinity and conformational stability of the sorafenib–FLT3 complex. Our results indicated that I863F is anticipated to destabilize the protein structure, possibly leading to functional changes in and implications for AML. While other I863 variants are anticipated to stabilize the protein structure, this could result in minimal functional changes. Nonetheless, the predicted detrimental impacts on the amino acid sequence (variants I836L, I836M, and I836S) suggest the possibility of unfavorable consequences for the protein’s functionality. The low binding affinity of I836L and I836S for sorafenib compared to the wild type indicates reduced binding affinities for the inhibitor (Figure 6). This suggests that the protein’s capacity to bind the inhibitor is diminished by these changes. Additionally, in protein biophysics, SASA refers to the surface area of a molecule that is accessible to the solvent molecules in its surroundings. Variations resulting in the SASA of variant proteins can provide insights into the structural and functional consequences that affect protein folding, stability, or conformational dynamics. Although this study lacks MD simulations, studying changing SASA effects is crucial for understanding and developing targeted therapeutic approaches. The increase in SASA of the I836L–sorafenib complex suggests that this variant may disrupt the protein–inhibitor binding interface, leading to a less stable interaction and resistance to inhibitors.

Numerous computational tools and methods are employed to forecast the effects of variants on the structure and function of proteins. Changes in the structure and function of proteins affect their binding to inhibitors. These computationally intensive approaches are valuable since they require less time and provide insights into how variants might impact a protein. Furthermore, the methodology used in the present study establishes an exhaustive workflow for variant annotation and can be used in the future for an infinite number of disease biomarker genes, emphasizing the value of bioinformatic methods in the investigation of genetic variations and their potential therapeutic implications. The limitation of our present integrated bioinformatics-based study is that the data used for SNVs are restricted to those available in public repositories only, and the variant effects are prediction-based only. In vivo validation is also missing, but that was beyond the scope and theme of the present study. An elaborate interaction study with all the known FLT3 inhibitors and their effects owing to each reported SNV in the gene can be planned for the future.

FLT3 mutations, including FLT3-ITD and FLT3-TKD, have been reported in around 30% of AML cases, and they are associated with a poor prognosis. Thus, multiple FLT3 inhibitors (midostaurin, lestaurtinib, sunitinib, sorafenib, semaxanib, and tandutinib) have been used to reduce the impact of FLT3 mutations and improve the overall survival of AML patients [46,47]. For example, midostaurin, an FLT3-inhibitor, has been approved by the FDA for treating AML, but it is a multi-kinase inhibitor that can interact with other kinases (PKC, VEGFR2, PDGFR-α/β, KIT, FGFR1), and it has not been specified whether it could be used in relapse and maintenance therapy or not. A system biology approach using whole-gene expression profiles to evaluate the potential benefits of the multi-target activity of potential FLT3 inhibitors for better understanding of their mechanism of action through molecular pathways and protein–protein interactions, as well as to check the treatment’s efficacy, i.e., their anti-leukemic effect on FLT3-mutated AML, seems promising [48,49,50,51,52]. For instance, a mathematical top–down systems biology approach based on machine learning and pattern recognition models that integrate all the available biological, pharmacological, and medical data has been carried out to simulate the behavior of human physiology in silico, and the models generated were centered around FLT3-mutated AML pathophysiology and the targets of midostaurin, daunorubicin, and cytarabine [53].

## 4. Materials and Methods

### 4.1. Association of FLT3 with AML

We searched the cancer OMICS data available on OncoDB (http://oncodb.org; accessed on 11 November 2023) [54] and the UALCAN portal (https://ualcan.path.uab.edu/; accessed on 17 November 2023) [55] to compare *FLT3* gene expression levels in cancer vs normal samples specific to each cancer type by generating box plots. Both of these portals use normalized RNA-Seq data to specifically correlate the expression of FLT3 in AML based on FLT3 gene mutational status.

### 4.2. Variant Annotation

In this study, bioinformatics tools were used to annotate variants of the FLT3 protein and predict deleterious effects and impacts on the function and structure of a protein using an in silico approach, as shown in Figure 7. The sequence of FLT3 was retrieved using UniProt database [P36888·FLT3_HUMAN] (accessed on 28 September 2023) [56]. Then, variants of AML were selected and annotated via the COSMIC database [57]. A number of distinct variants of FLT3 were also scanned from MOKCa (Mutations, Oncogenes, Knowledge, and Cancer) (http://strubiol.icr.ac.uk/extra/mokca; accessed on 14 February 2024) to gain an idea about the frequencies reported for each variant [58].

### 4.3. Variant Deleterious Effects

To predict the possible impact of mutagenesis variants, we analyzed the amino acid sequence (AA) in FASTA format using numerous tools, including Sorting Intolerant from Tolerant (SIFT) [59] to predict whether protein function is affected by amino acid substitutions (>0.05 = tolerated and <0.05 = deleterious); PolyPhen-2 (Polymorphism Phenotyping v2) [60] to predict the probable impact of an amino acid substitution on protein structure and function (0 = probably damaging, 1 = possibly damaging, 2 = benign, 3 = unknown); and the PredictSNP server [61] (neutral and deleterious). Prediction of disease association of deleterious amino acid polymorphisms and protein functional annotation was carried out using SNPs&GO based on a support-vector machine approach [62] (http://snps.biofold.org/snps-and-go; accessed on 26 November 2023).

### 4.4. FLT3 Protein and Its Re-Modeling

The three-dimensional (3D) structure of wild-type (WT) FLT3 was obtained from the RCSB Protein Data Bank (ID: 1rjb), and InterProScan (accessed on 30 November 2023) [63] was used for functional analysis of FLT3 sequences, which were then visualized using Schrodinger’s PyMOL. The missing residues in 1rjb were built using a Swiss model based on the target sequence of the FLT3 protein. Residue I836 was mutated (individually into 836F, 836L, 836M, 836S, and 836V) using PyMOL, and then the WT and mutant FLT3 protein 3D structures were assessed through the ProSA web server [64].

### 4.5. Protein Stability Analysis for the Mutation Hotspot

Missense3D [65,66] was used to predict the impact of missense variants on protein interfaces using 3D. To analyze mutant protein dynamics and determine the effects of mutations on protein stability, DynaMut [67] was used. A negative ΔΔG value indicates that the mutation is predicted to be destabilizing, and a positive value is predicted to be stabilizing. Protein stability differences for structurally conserved mutations between the WT and mutants were computed using CUPSAT [68]. Protein kinase stability may be impacted by cancer mutations that have a functional role, as indicated by negative values of protein stability alterations that correlate with destabilizing mutations [33].

### 4.6. Molecular Docking

To understand how the wild-type and I836 mutants interacted with the FLT3 inhibitors, molecular docking was carried out using AutoDockTools 1.5.6. Following preparation of the WT, I836F, I836L, I836M, I836S, and I836V proteins, they were docked to sorafenib, which was retrieved from PubChem (CID: 216239). The best pose was chosen based on the lowest binding energy and then visualized using Discovery Studio V21.1.0. Solvent-accessible surface area (SASA) was also computed using Discovery Studio Visualizer V21.1.0.

## 5. Conclusions

The structure-based functional annotation and prediction of cancer variations’ impact on biomarker proteins, including FLT3, will help in better understanding the molecular pathology of tumorigenesis and drug-resistance mechanisms. Inspiring insights into the variant landscape can pave the way for innovative and effective AML therapies that can overcome the resistance to present FLT3 inhibitors. An analogous plan can be followed in the future for other gene mutations and variations associated with AML and other cancers.

## Figures and Tables

**Figure 1 ijms-25-03419-f001:**
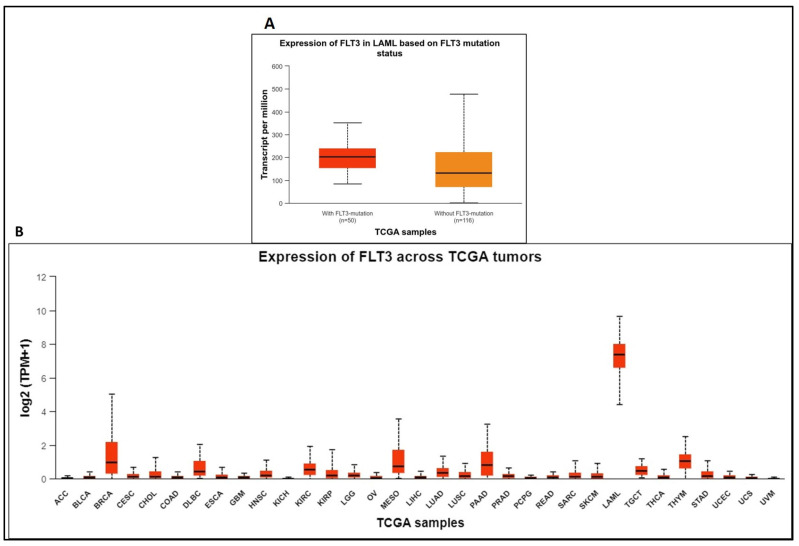
(**A**). UALCAN database correlates FLT3 mutations with AML. (**B**). Expression levels of FLT3 across different TCGA tumors.

**Figure 2 ijms-25-03419-f002:**
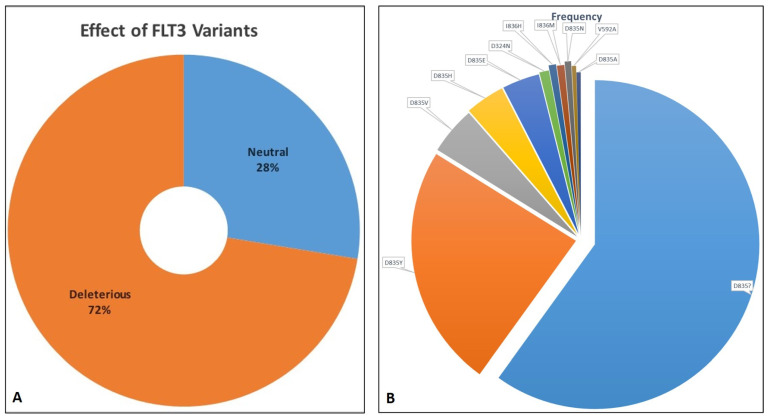
(**A**). Distribution of FLT3 variants as deleterious and neutral. (**B**). Distribution of distinct FLT3 variants based on their frequency as retrieved from MoKCa database.

**Figure 3 ijms-25-03419-f003:**
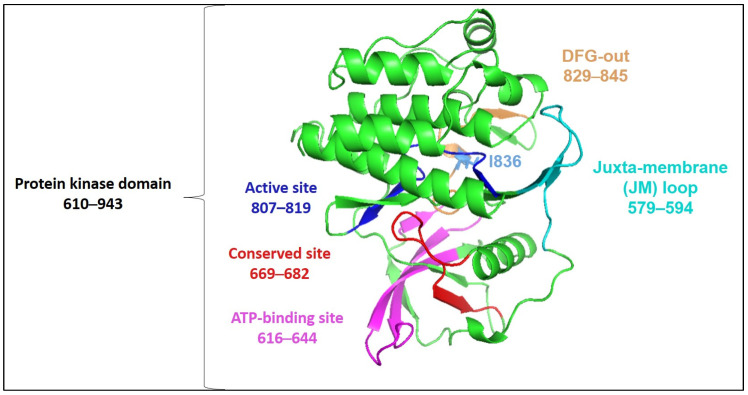
Three-dimensional structure of FLT3 kinase (based on PDB 1RJB) annotated by InterProScan (accessed on 30 November 2023).

**Figure 4 ijms-25-03419-f004:**
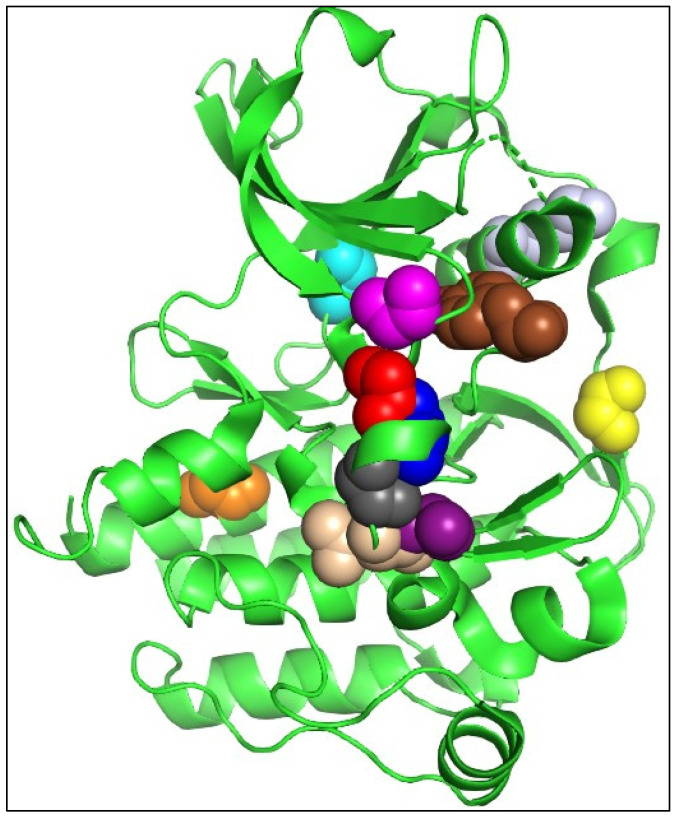
Selected mutational hotspots of FLT3 are shown as colored spheres (Y572 as brown, Y591 as yellow, G619 as magenta, K663 as light grey, N676 as cyan, F691 as orange, D835 as red, I836 as blue, D839 as dark grey, N841 as beige, and Y842 as purple). Figure was drawn using PyMOL 2.3.3.

**Figure 5 ijms-25-03419-f005:**
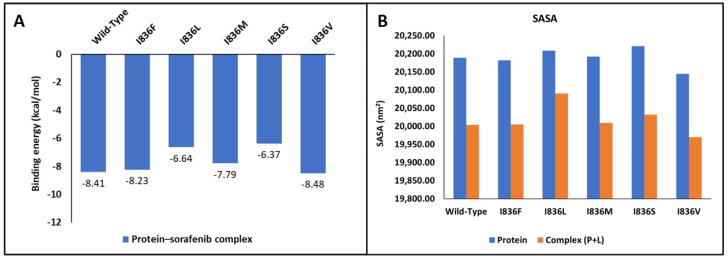
Molecular docking analysis. (**A**) Binding energy of interaction between wild-type and FLT3’s I836 mutants with its inhibitor. (**B**) SASA analysis of proteins.

**Figure 6 ijms-25-03419-f006:**
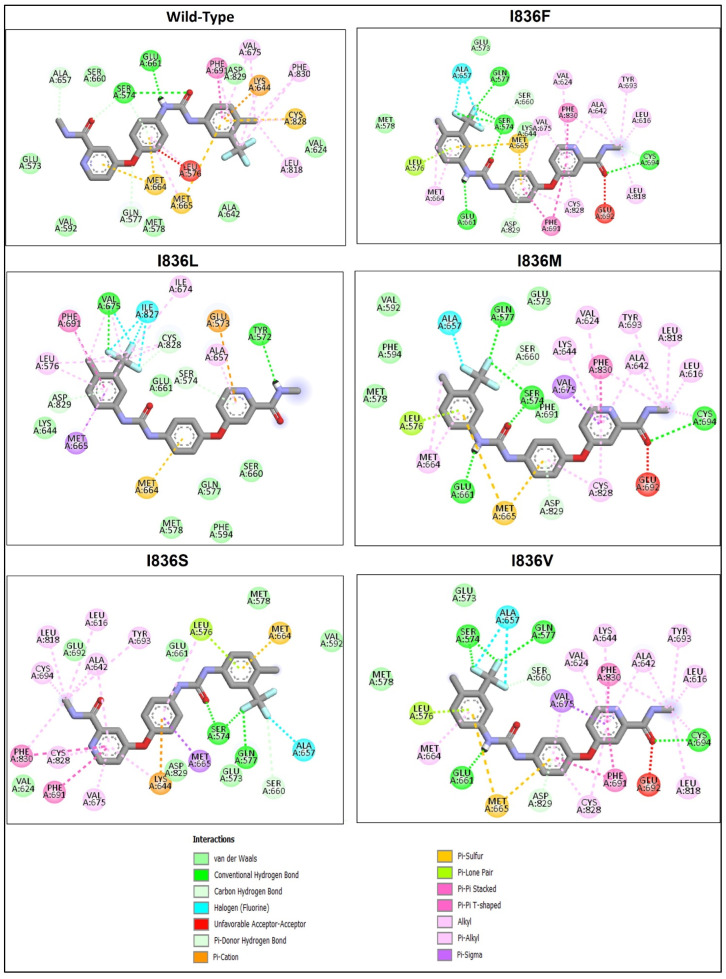
Analysis of the residue interaction between FLT3’s WT and I836 mutant with FLT3 inhibitor sorafenib.

**Figure 7 ijms-25-03419-f007:**
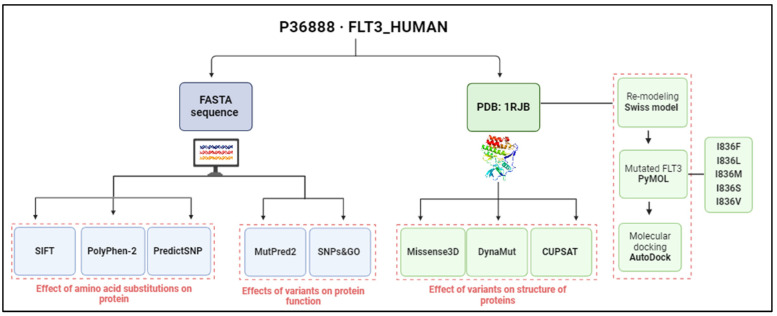
Overview of workflow for profiling FLT3’s variants and their effects. Analysis based on sequence is shown in blue, whereas the structure-based one is shown in green (image created using BioRender.com).

**Table 1 ijms-25-03419-t001:** Variants and effects of amino acid substitutions on protein.

Variants ID	Mutation	Mutation Type	SIFT	Polyphen-2	PredictSNP	
					Effect	Confidence
RCV000444818	Y572C	Substitution—Missense	0	1.000	Deleterious	87%
RCV000445102	V579A	Substitution—Missense	0.01	0.551	Neutral	63%
RCV000420236	Y591C	Substitution—Missense	0	1.000	Deleterious	72%
RCV000441431	Y591D	Substitution—Missense	0	0.996	Deleterious	55%
RCV000435462	V592A	Substitution—Missense	0	0.742	Neutral	63%
RCV000432251	F594L	Substitution—Missense	0.48	0.999	Neutral	68%
RCV000437384	G619C	Insertion—In frame	0	1.000	Deleterious	87%
RCV000426662	D651G	Substitution—Missense	0.23	0.326	Neutral	83%
RCV000422333	K663Q	Substitution—Missense	0	0.981	Deleterious	51%
RCV000427705	N676K	Substitution—Missense	0	1.000	Deleterious	72%
RCV000443196	I687F	Substitution—Missense	0.05	0.010	Neutral	63%
RCV000420978	F691I	Substitution—Missense	0	0.881	Deleterious	61%
RCV000444069	D835A	Substitution—Missense	0	1.000	Deleterious	87%
RCV000424615	D835E	Substitution—Missense	0	0.959	Deleterious	87%
RCV000017663	D835F	Substitution—Missense	0	0.995	Deleterious	87%
RCV000017662	D835H	Substitution—Missense	0	1.000	Deleterious	72%
RCV000017663	D835N	Substitution—Missense	0	0.938	Deleterious	61%
RCV000017660	D835V	Substitution—Missense	0	0.999	Deleterious	87%
RCV000017665	D835Y	Substitution—Missense	0	1.000	Deleterious	87%
RCV000417837	I836F	Substitution—Missense	0	0.991	Deleterious	61%
RCV000432941	I836L	Substitution—Missense	0	0.204	Neutral	75%
RCV000422249	I836M	Substitution—Missense	0	1.000	Deleterious	61%
RCV000444162	I836S	Substitution—Missense	0	1.000	Deleterious	76%
RCV000428691	I836V	Substitution—Missense	0	0.230	Neutral	83%
RCV000429280	D839G	Substitution—Missense	0	0.880	Neutral	61%
RCV000440005	N841H	Substitution—Missense	0	0.022	Deleterious	51%
RCV000427616	N841K	Substitution—Missense	0.01	0.246	Deleterious	61%
RCV000421989	Y842C	Substitution—Missense	0	1.000	Deleterious	87%
RCV000431811	Y842H	Substitution—Missense	0	1.000	Deleterious	76%

**Table 2 ijms-25-03419-t002:** Effects of variants on protein function.

Variants	MutPred2			SNPs&GO	
	Score	Molecular Mechanisms	*p*-Values	Effect	Reliability Index
Y572C	0.66	Altered transmembrane protein	6.00 × 10^−3^	Neutral	2
V579A	0.37	-	-	Neutral	5
Y591C	0.69	Altered ordered interface Loss of phosphorylation at Y591 Loss of sulfation at Y591 Altered transmembrane protein	3.6 × 10^−3^ 0.02 4.7 × 10^−4^ 3.1 × 10^−3^	Disease	5
Y591D	0.85	Altered ordered interface Gain of relative solvent accessibility Loss of phosphorylation at Y591 Loss of sulfation at Y591 Altered transmembrane protein	9.0 × 10^−3^ 0.01 0.02 4.7 × 10^−4^ 3.9 × 10^−3^	Disease	5
V592A	0.27	-	-	Neutral	4
F594L	0.44	-	-	Neutral	1
G619C	0.91	Loss of acetylation at K614 Gain of relative solvent accessibility Loss of methylation at K623 Altered transmembrane protein	7.5 × 10^−3^ 0.03 0.01 0.03	Disease	5
D651G	0.31	-	-	Neutral	3
K663Q	0.46	-	-	Neutral	7
N676K	0.89	Gain of helix Altered transmembrane protein	0.05 0.04	Disease	0
I687F	0.69	Altered transmembrane protein	0.02	Neutral	0
F691I	0.76	-	-	Disease	2
D835A	0.85	Loss of relative solvent accessibility Altered ordered interface Loss of loop Loss of allosteric site at R834 Altered transmembrane protein Altered metal binding Altered DNA binding	8.3 × 10^−3^ 0.05 0.03 8.6 × 10^−3^ 1.5 × 10^−3^ 0.03 0.04	Disease	3
D835E	0.70	Loss of allosteric site at R834 Loss of relative solvent accessibility Altered transmembrane protein Altered metal binding Altered DNA binding	0.01 0.03 2.9 × 10^−3^ 0.05 0.05	Neutral	0
D835F	0.9	Loss of relative solvent accessibility Loss of allosteric site at R834 Altered ordered interface Altered transmembrane protein Altered metal binding Altered DNA binding	4.1 × 10^−3^ 5.3 × 10^−3^ 0.04 1.1 × 10^−3^ 0.03 0.04	Disease	7
D835H	0.87	Altered metal binding Loss of relative solvent accessibility Altered ordered interface Loss of loop Loss of allosteric site at R834 Altered transmembrane protein Altered DNA binding	0.01 0.01 0.04 0.04 9.9 × 10^−3^ 1.3 × 10^−3^ 0.04	Disease	4
D835N	0.75	Altered ordered interface Loss of loop Loss of relative solvent accessibility Gain of allosteric site at R834 Altered transmembrane protein Altered metal binding Altered DNA binding	0.02 0.03 0.03 8.5 × 10^−3^ 2.6 × 10^−3^ 0.05 0.04	Disease	0
D835V	0.87	Loss of relative solvent accessibility Altered ordered interface Loss of loop Loss of allosteric site at R834 Altered transmembrane protein Altered metal binding Altered DNA binding	4.3 × 10^−3^ 0.03 0.02 8.5 × 10^−3^ 7.7 × 10^−4^ 0.03 0.04	Disease	6
D835Y	0.90	Loss of relative solvent accessibility Loss of allosteric site at R834 Loss of loop Altered transmembrane protein Altered metal binding Altered DNA binding	8.7 × 10^−3^ 7.8 × 10^−3^ 0.04 1.4 × 10^−3^ 0.03 0.04	Disease	6
I836F	0.80	Gain of allosteric site at R834 Gain of loop Altered transmembrane protein Gain of relative solvent accessibility Altered DNA binding Altered metal binding Gain of proteolytic cleavage at D835	3.3 × 10^−4^ 0.02 1.5 × 10^−3^ 0.04 0.04 0.02 0.04	Disease	2
I836L	0.54	Gain of allosteric site at R834 Altered ordered interface Gain of relative solvent accessibility Altered transmembrane protein Altered metal binding Altered DNA binding Gain of proteolytic cleavage at D835	1.3 × 10^−3^ 0.02 0.04 2.8 × 10^−3^ 0.02 0.04 0.04	Neutral	4
I836M	0.63	Gain of allosteric site at R834 Gain of relative solvent accessibility Altered ordered interface Altered transmembrane protein Altered metal binding Altered DNA binding	4.0 × 10^−3^ 0.03 0.05 2.7 × 10^−3^ 0.05 0.04	Neutral	3
I836S	0.89	Gain of relative solvent accessibility Altered ordered interface Gain of allosteric site at R834 Gain of loop Gain of B-factor Altered transmembrane protein Altered DNA binding Altered metal binding Gain of proteolytic cleavage at D835 Altered stability	6.3 × 10^−3^ 0.02 2.5 × 10^−3^ 0.03 0.02 2.2 × 10^−3^ 0.02 0.05 9.6 × 10^−3^ 0.03	Disease	6
I836V	0.30	-	-	Neutral	8
D839G	0.871	Altered ordered interface Loss of relative solvent accessibility Loss of allosteric site at R834 Altered transmembrane protein Altered metal binding Altered DNA binding	0.03 0.02 6.3 × 10^−3^ 1.5 × 10^−3^ 0.04 0.05	Disease	3
N841H	0.453	-	-	Neutral	3
N841K	0.622	Gain of acetylation at N841 Altered ordered interface Loss of relative solvent accessibility Altered transmembrane protein Altered metal binding Gain of ubiquitylation at N841 Altered stability	9.8 × 10^−4^ 0.04 0.04 2.0 × 10^−3^ 0.05 0.02 0.03	Neutral	2
Y842C	0.859	Altered ordered interface Altered transmembrane protein Loss of relative solvent accessibility Loss of strand Altered metal binding Gain of disulfide linkage at Y842	6.3 × 10^−4^ 7.1 × 10^−4^ 0.03 0.04 0.04 0.04	Disease	2
Y842H	0.823	Altered ordered interface Gain of relative solvent accessibility Altered transmembrane protein Altered DNA binding Altered metal binding Altered stability	1.2 × 10^−3^ 0.01 1.2 × 10^−3^ 0.03 0.03 0.03	Disease	7

**Table 3 ijms-25-03419-t003:** Effect of variants on the structure of proteins.

Variants	Missense3D	DynaMut-Predicted ΔΔG (kcal/mol)	CUPSAT-predicted ΔΔG (kcal/mol)
Y572C	Structural damage detected Cavity altered	−0.936 kcal/mol	6.96
V579A	Structural damage detected Buried H-bond breakage	−1.173 kcal/mol	2.06
Y591C	No structural damage detected	−0.984 kcal/mol	-1.0
Y591D	No structural damage detected	−1.325 kcal/mol	2.98
V592A	No structural damage detected	−1.41 kcal/mol	3.67
F594L	Structural damage detected Buried H-bond breakage	−1.432 kcal/mol	1.73
G619C	Structural damage detected Buried Gly replaced Buried/exposed switch	0.076 kcal/mol	-2.78
D651G	No structural damage detected	−0.071 kcal/mol	-
K663Q	Structural damage detected Buried charge replaced	−0.287 kcal/mol	0.25
N676K	Structural damage detected Buried charge introduced	0.736 kcal/mol	3.28
I687F	Structural damage detected Buried/exposed switch	1.402 kcal/mol	1.35
F691I	No structural damage detected	−0.048 kcal/mol	4.43
D835A	No structural damage detected	−0.531 kcal/mol	0.76
D835E	No structural damage detected	−0.164 kcal/mol	1.1
D835F	No structural damage detected	1.538 kcal/mol	−0.44
D835H	No structural damage detected	0.029 kcal/mol	0.0
D835N	No structural damage detected	−0.0 kcal/mol	0.01
D835V	No structural damage detected	0.77 kcal/mol	−0.04
D835Y	No structural damage detected	0.041 kcal/mol	0.75
I836F	Structural damage detected Cavity altered	1.105 kcal/mol	−0.29
I836L	No structural damage detected	0.137 kcal/mol	0.6
I836M	No structural damage detected	0.429 kcal/mol	3.17
I836S	No structural damage detected	−1.449 kcal/mol	2.44
I836V	No structural damage detected	−0.183 kcal/mol	1.55
D839G	No structural damage detected	−0.922 kcal/mol	4.06
N841H	No structural damage detected	1.904 kcal/mol	1.45
N841K	No structural damage detected	1.669 kcal/mol	1.79
Y842C	No structural damage detected	−1.037 kcal/mol	2.37
Y842H	No structural damage detected	−1.224 kcal/mol	4.99

## Data Availability

All the data were retrieved from the public domain, and links are mentioned in the manuscript.

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
