# Peer review of "Landscape of FLT3 Variations Associated with Structural and Functional Impact on Acute Myeloid Leukemia: A Computational Study"

_ijms, 2024, doi:10.3390/ijms25063419_

Round 1

Reviewer 1 Report

Comments and Suggestions for Authors

The Authors provide an interesting paper about FLT3 mutations in AML.

In the Abstract, a 5-year survival of 29.5% is quoted for AML, from the reference of older patients; so this quote should be refined (pointing out the difference between older and younger patients, HSCT and so on).

Also, in the Introduction there should be a mention of the specific mutations resistant to FLT3 inhibitors.

Reviewer 2 Report

Comments and Suggestions for Authors

Miza Z. and coauthors in this article analyzed the impact of fms-like tyrosine kinase 3 (FLT3) variants in Acute myeloid leukemia (AML) pathogenesis by integrating information from different databases and resources available online.

Mutations in FLT3 gene occur in 30% of AML cases and are constitutively activating, leading to ligand-independent FLT3 signaling and cellular proliferation.

They analyzed in silico twenty-nine FLT3 variants to explore their potential impact on protein structure and biological functions. Using different tools, they found that single nucleotide variations of FLT3 may have impact on its structure and function and consequently could be predicted their pathogenetic impact or a low binding affinity to inhibitors usable in targeted therapies.

The data are potentially interesting, but I found their presentation often not very organized and not easy to understand. The introduction should better present the background and objectives of this study. The results are often not presented clearly and the legends to the figures are also not very explanatory. Finally, the discussion must highlight the significance of the results.

Major comments:

1)    Introduction lane 50-51 “Mutations in the fms-like tyrosine kinase 3 (FLT3) gene are linked with poor prognosis and overexpressed on most AML blasts “.

Currently not all FLT3 mutations have a clear prognostic relevance especially the FLT3-TKD mutations that are point mutations within the receptor’s activation loop. It would be appropriate for the authors to clarify on this topic. In addition the authors often take for granted important information to understand the goal of this work. The introduction should be revised.

2)    Results obtained by interrogating the OncoDB database on the differential expression of FLT3 in AML versus normal samples appear to be completely wrong. In fact, normal controls for AML samples are not available in the OncoDB database as also shown in fig 2A (normal n=0).

3)    It is not clear what the authors want to highlight in fig 2B. Perhaps that there is no significant difference in FLT3 expression between FLT3 mutated and non-mutated AML cases? But in this context I miss the point of specifying this result.

4)    The overexpression of FLT3 compared to other types of tumors shown in fig 2C is correct, but since normal comparators are missing, in my opinion it is useless for the purposes of the work.

5)    The experiments of molecular docking and interaction analysis they are interesting also as they are the only experimental validation data of this paper. However, they are presented in an unorganized and difficult to understand manner. Furthermore, the data in figure 6 are discussed before those in figure 5.

6)    The discussion is mainly another introduction. Only the last part is the actual discussion of the data.

Comments on the Quality of English Language

The quality of the English is quite good, but the authors have to make an effort to make the text more understandable to readers who are not specialized in these studies.

Reviewer 3 Report

Comments and Suggestions for Authors

Although the main focus of the research aligns with the objectives of the journal, some concerns need to be addressed before publication. One of these concerns is particularly crucial, and therefore, the article requires major revision.
1-In Figure 6, only the interaction of WT and I836 mutant FLT3 with sorafenib has been presented. However, the introduction mentions several types of FLT3 inhibitors. Therefore, it is necessary to evaluate the interaction of WT and I836 mutant FLT3 with other types of mutation inhibitors, including at least one drug from each group of inhibitors.
2-The discussion section is brief, and the study's outcomes have not been compared adequately with similar studies. It is highly recommended to improve this section.
3-The limitations of the study and the perspective of related future works were not adequately addressed in the discussion section.
4-The introduction does not adequately describe the novelty of the study and the gap between the studies that led to the design of this study.
5- It is worth noting that the Wnt/β-catenin pathway can modulate the sensitivity of mutant FLT3 receptor kinase inhibitors. Additionally, the inhibition of glycogen synthase kinase 3 can affect FLT3 mutations, and lithium can be categorized as one of the FLT3 inhibitors. Therefore, it is recommended to read and cite the following sources:

    https://journals.sagepub.com/doi/full/10.1177/1947601910362446
    https://link.springer.com/article/10.1007/s12032-022-01899-2
    https://www.sciencedirect.com/science/article/abs/pii/S2212492617300866
    https://link.springer.com/article/10.1007/s10924-022-02615-x
    https://www.mdpi.com/2079-4983/13/4/162
    https://www.frontiersin.org/articles/10.3389/fnmol.2022.1028963/full
6-Please elaborate on how this study outcome could be beneficial along with system biology results. you can see:
https://www.mdpi.com/2673-7426/2/3/24

In summary, the article requires major revision to address the concerns mentioned above before it can be considered for publication.

Round 2

Reviewer 2 Report

Comments and Suggestions for Authors

The authors responded to all my comments, making both the introduction and the discussion of the manuscript much clearer and more effective. Changes in the results section are also appropriate.

I only have one minor comment: in paragraph 3.5 line 281 "figure 6" I think refers to figure 7.

Author Response

Thanks for the kind appreciation. Correction done.